# Combinatorial Inference against Label Noise

**Paul Hongsuck Seo**[†‡]
†Computer Vision Lab.
POSTECH
hsseo@postech.ac.kr

**Geeho Kim**[‡]          **Bohyung Han**[‡]
‡Computer Vision Lab. & ASRI
Seoul National University
{snow1234, bhhan}@snu.ac.kr

## Abstract

Label noise is one of the critical sources that degrade generalization performance of deep neural networks significantly. To handle the label noise issue in a principled way, we propose a unique classification framework of constructing multiple models in heterogeneous coarse-grained meta-class spaces and making joint inference of the trained models for the final predictions in the original (base) class space. Our approach reduces noise level by simply constructing meta-classes and improves accuracy via combinatorial inferences over multiple constituent classifiers. Since the proposed framework has distinct and complementary properties for the given problem, we can even incorporate additional off-the-shelf learning algorithms to improve accuracy further. We also introduce techniques to organize multiple heterogeneous meta-class sets using $k$-means clustering and identify a desirable subset leading to learn compact models. Our extensive experiments demonstrate outstanding performance in terms of accuracy and efficiency compared to the state-of-the-art methods under various synthetic noise configurations and in a real-world noisy dataset.

## 1   Introduction

Construction of a large-scale dataset is labor-intensive and time-consuming, which makes it inevitable to introduce a substantial level of label noise and inconsistency. This issue is aggravated if the data collection relies on crowd-sourcing [1, 2] or internet search engines [3, 4] without proper curation. More importantly, many real-world problems inherently involve a significant amount of noise and it is extremely important to train machine learning models that can handle such a challenge effectively. Figure 1 presents several noisy examples in WebVision benchmark [4], where training examples are collected from Google and Flickr by feeding queries corresponding to the ImageNet class labels. Although a moderate level of noise is sometimes useful for regularization, label noise is a critical source of underfitting or overfitting. In the case of deep neural networks, models can easily memorize a large number of noisy labels and, consequently, are prone to degrade their generalization performance [5].

To tackle various kinds of label noise, existing mainstream approaches either attempt to filter noisy examples out [6–11] or correct noisy labels based on the network predictions [12–15]. However, these methods are similar to solving the chicken-and-egg problem and their unstable on-the-fly noise recognition process may result in poor generalization performance. On the other hand, [16–21] assume the correct labels to be latent and learn the networks inferring the latent correct labels by estimating noise transition matrices. Although these methods estimate the noise injection process directly, they are also suboptimal when the network is capable of adapting to label noise as discussed in [10].

While the prior approaches typically focus on developing noise-resistant training algorithms given noise levels, our algorithm takes a totally different perspective of reducing noise level and learning a

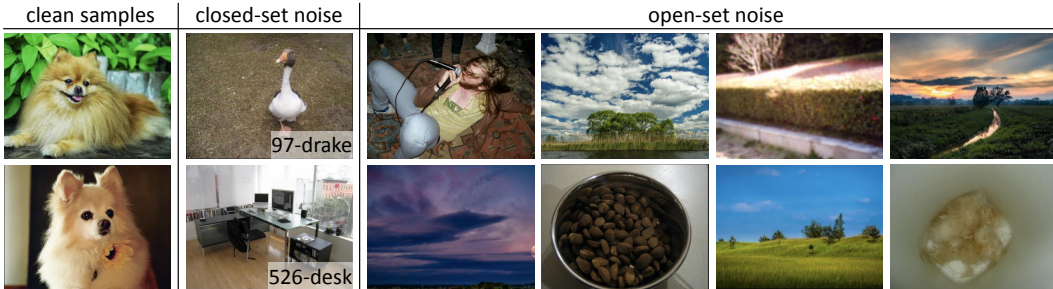

Figure 1: Examples of `259-pomeranian` class in WebVision [4]. In addition to clean samples, the dataset contains closed- and open-set noises, where the examples with closed-set noise are mislabeled with known classes while the ones with open-set noise are associated with unknown class labels.

representation robust to label noise. In our framework, we automatically generate multiple coarse-grained meta-class sets, each of which constructs a heterogeneous partition of the original class set. Then, we train classifiers on individual meta-class sets and make the final prediction using an output combination of the classifiers. Note that the combination process allows us to uniquely identify the original classes despite the coarse representation learning on meta-class spaces. Learning on meta-class spaces actually reduces the level of label noise because multiple classes in an original class space collapse to a single meta-class and the label noise within the same meta-class becomes invisible on the meta-class space.

The contribution of this paper is three-fold; (1) we successfully reduce the amount of label noise by constructing meta-classes of multiple base classes; (2) we propose a novel combinatorial classification framework, where inference on the original class space is given by combining the predictions on multiple meta-class spaces; (3) we demonstrate the robustness of the proposed method through extensive controlled experiments as well as the evaluation on a real-world dataset with label noise.

The rest of this paper is organized as follows. Section 2 reviews previous approaches against datasets with label noise and other related techniques. Then, we formally describe the proposed compositional classification method, and demonstrate the effectiveness of our method in Section 3 and 4, respectively. Finally, we conclude our paper in Section 5.

## 2   Related Work

One common approach to learning with noisy data is to correct or filter out noisy examples during training [6–15, 22–25]. Existing methods adopt their own criteria to identify the noisy samples. There exist several techniques to employ the confidence scores of models as the signal of noise in [11–14] while [8] incorporates a contrastive loss term to iteratively identify noisy samples. Deep bilevel learning [9] attempts to find reliable mini-batches based on the distances between the mini-batches in training and validation datasets. Multiple networks have often been adopted to identify noisy examples. For example, two networks with an identical architecture are jointly trained to identify noisy samples in each batch [6, 11] whereas a separate teacher network is employed to select samples for training a student network. Contrary to the approaches making hard decisions on noisy sample selection, there are a handful of algorithms relying on the soft penalization of potentially noisy examples by designing noise-robust loss functions [10, 23], using knowledge distillation [24] and adding regularizers [22]. Although these methods are often motivated by intuitive understanding of classification models, their ad-hoc procedures often lack theoretical support and hamper reproducibility.

Another line of methods estimates a noise transition matrix capturing transition probabilities from correct labels to corrupted ones [16–21]. Some of them [16–18] adopt the standard backpropagation to estimate the transition matrix and train the network simultaneously while a pretrained network is often used for the transition matrix estimation [19]. To improve the quality of the estimated transition matrices, additional clean data [21] or manually defined constraints [20] are sometimes integrated during the matrix estimation process.

Although all these existing approaches cover various aspects of training with noisy data, they typically assume that the noise-level of a dataset is irrevocable and therefore focus on developing algorithms that

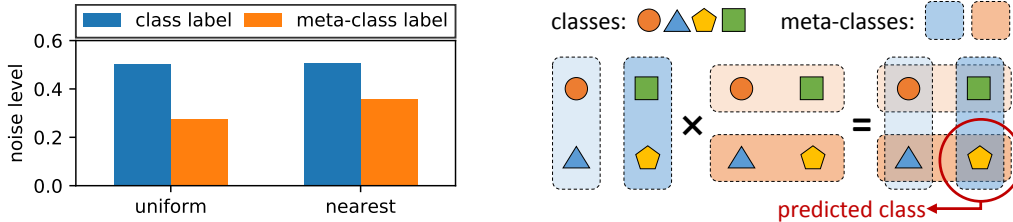

Figure 2: Motivation and concept of combinatorial classification. (left) Empirical noise-level reduction by use of meta-class labels on CUB-200 with closed-set noise; the noise rates in the meta-class level show the average of all meta-class sets. (right) Illustration of combinatorial classification with two binary meta-class sets on four original classes. By combining the coarse-grained meta-classes, it is possible to predict fine-grained original class labels.

avoid overfitting to a noisy training set by identifying noisy examples or modeling noise distribution. In contrast, we propose a novel output representation method that directly reduces the noise-level of a given dataset, and a model that predicts class labels based on the proposed representations.

The proposed combinatorial classification solves the target problem by combining solutions of multiple subproblems encoded by class codewords and there are several related methods in this aspect. Product quantization [27, 28] measures distances in multiple quantized subspaces and combines them to perform the approximate nearest neighbor search in the original space. A recently proposed an image geolocalization technique by classification achieves the fine-grained quantization by combining multiple coarse-grained classifications [29], while a similar approach is proposed for metric learning for retrieval tasks [30]. Unlike these works targeting regression or retrieval tasks on continuous spaces, our approach deals with a classification problem on a discrete output space. Ensemble methods [31–37] also have the similar concept to our algorithm but are different in the sense that their constituent models have the common output space. One of the most closely related work is the classification by error-correcting output code [26]. This technique combines the results of binary classifiers to solve multi-class classification problems and proposes deterministic processes to generate and predict the binarized codewords based on Hamming distance. In contrast, we generate codewords by exploiting the semantics of the original classes and combine the predicted scores to construct the compositional classifier robust to label noise.

## 3 Combinatorial Classification

### 3.1 Class Codewords

As in the ordinary classification, our goal is to predict a class label $y \in \mathcal{C}$ given an input $x$, where $\mathcal{C} = \{c_1, \ldots, c_K\}$ is a set of $K$ disjoint classes. Unlike conventional classification approaches that directly predict the output class $y$, our model estimates $y$ by predicting its corresponding unique codeword. To construct the class codewords, we define $M$ meta-class sets, each of which is given by a unique partitioning of $\mathcal{C}$. Specifically, each meta-class set denoted by $\mathcal{C}^m$ ($m = 1, \ldots, M$) has $K'(\ll K)$ meta-classes, *i.e.*, $\mathcal{C}^m = \{c_1^m, \ldots, c_{K'}^m\}$, where multiple original classes are merged into a single meta-class, which results in a coarse-grained class definition. Then, each class $c_i$ is represented by a $M$-ary codeword, $c_{i_1}^1 c_{i_2}^2 \ldots c_{i_M}^M$, where $c_{i_m}^m$ corresponds to a meta-class to which $c_i$ belongs in a meta-class set $\mathcal{C}^m$.

When training data have label noise, classification on a coarse-grained meta-class set naturally reduces noise level of the dataset. Formally, let $\eta(\hat{\mathcal{D}})$ be the noise level of a dataset $\hat{\mathcal{D}} = \{(x_i, \hat{y}_i)\}_{i=1}^{N}$, which is given by

$$\eta(\hat{\mathcal{D}}) = \mathbb{E}_{\hat{\mathcal{D}}}[\mathbb{1}(y \neq \hat{y})] = \frac{1}{N} \sum_{i=1}^{N} \mathbb{1}(y_i \neq \hat{y}_i), \qquad (1)$$

where $\hat{y}_i$ means a label potentially corrupted from a clean label $y_i$ and $\mathbb{1}$ is an indicator function. Although two examples $x_i$ and $x_j$ belong to the same class but the label of $x_j$ is corrupted from $y_j(= y_i)$ to $\hat{y}_j$, the two classes corresponding to $y_i$ and $\hat{y}_j$ can be merged into the same meta-class, which removes the label noise in the meta-class level. Consequently, the noise level with the meta-

class representations is lower than that with the original class space, *i.e.*, $\eta(\hat{\mathcal{D}}^m) \leq \eta(\hat{\mathcal{D}})$, where $\hat{\mathcal{D}}^m = (x_i, \hat{y}_i^m)_{i=1}^N$ is the dataset associated with meta-class labels in $\mathcal{C}^m$. In Figure 2(left), we make empirical observations of the noise-level reduction on CUB-200 with two different noise injection schemes[1]. The noise levels are significantly reduced regardless of noise-types by converting the original class spaces into meta-class representations.

Although a coarse-grained meta-class representation reduces noise level, it is not capable of distinguishing the base classes in the original class space $\mathcal{C}$. We resolve this limitation by introducing multiple heterogeneous meta-class sets and exploiting their compositions. Even if multiple classes are collapsed to a single meta-class within a meta-class set, it is possible to provide a unique class codeword to each of the original classes by using a sufficiently large number of meta-class sets. In this way, we convert noisy labels in $\hat{\mathcal{D}}$ to partially noisy codewords.

### 3.2 Classification with Class Codewords

Given noise-robust class codewords defined above, we now discuss the classification method that predicts the class codewords to identify the class label $y$. Unlike ordinary classifiers directly predicting the class label on $\mathcal{C}$, we construct $M$ constituent classifiers, each of which estimates a distribution on a meta-class set $\mathcal{C}^m$ ($m = 1, \ldots, M$), and combine their predictions to obtain class labels on $\mathcal{C}$. This process is referred to as combinatorial classification and is illustrated in Figure 2(right).

**Inference**  A constituent classifier estimates the conditional distribution $P(c_k^m|x)$ on a meta-class set $\mathcal{C}^m$. Given $M$ constituent classifiers, we obtain the conditional probability of $c_k \in \mathcal{C}$ by combining the predictions of constituent classifiers as follows:

$$P(c_k|x) = \frac{\prod_{m=1}^M P\left(\mathrm{meta}(c_k; m)|x\right)}{\sum_{j=1}^K \prod_{m=1}^M P\left(\mathrm{meta}(c_j; m)|x\right)}, \tag{2}$$

where $\mathrm{meta}(c_k; m)$ returns the meta-class label containing the base class $c_k$ in the $m$-th meta-class set. The denominator in Eq. (2) is the normalization term deriving $\sum_{k=1}^K P(c_k|x) = 1$.

**Training**  We train our model by minimizing the sum of negative log-likelihoods with respect to the ground-truth meta-class labels $\mathrm{meta}(y; m)$ that contain the ground-truth label $y \in \mathcal{C}$, *i.e.*,

$$-\sum_{m=1}^M \log P(\mathrm{meta}(y; m)|x). \tag{3}$$

This objective encourages the constituent classifiers to maximize the prediction scores of the true meta-classes. Our algorithm employs the objective in Eq. (3) even though the following objective, minimizing the negative log-likelihood of the ground-truth class label $y$, is also a reasonable option:

$$-\log P(y|x) = -\sum_{m=1}^M \log P\left(\mathrm{meta}\left(y; m\right)|x\right) + \log \sum_{k=1}^K \prod_{m=1}^M P(\mathrm{meta}(c_k; m)|x). \tag{4}$$

Although this objective is directly related to the inference procedure in Eq. (2), it turns out to be not effective. Note that the second term of the right hand side in this equation corresponds to the denominator in Eq. (2), and penalizes the scores of the classes other than the true one, *i.e.*, $\mathcal{C} \setminus \{y\}$. Since a ground truth meta-class may contain non-ground-truth original class labels, the penalty given to these non-ground-truth class labels can be propagated to the ground-truth meta-classes. Consequently, the optimization of each constituent classifier becomes more challenging.

**Deep combinatorial classifier**  We implement our model using a deep neural network with a shared feature extractor and $M$ parallel branches corresponding to the individual constituent classifiers. Since the shared feature extractor receives the supervisory signal from all the $M$ classifiers, we scale down the gradients of the shared feature extractor by a factor of $M$ for backpropagation. Note that our approach uses the exactly same number of parameters with a flat classifier in the feature extractor and its model size is rather smaller in total even with multiple network branches. This is mainly because the number of meta-classes is much less than the number of base classes ($K' \ll K$) and, consequently, each classifier requires fewer parameters.

### 3.3 Configuring Meta-class Sets

To implement our method, one needs to define heterogeneous meta-class sets. A naive approach for determining a meta-class set configuration is to randomly assign each class to one of meta-classes in a meta-class set. However, this method may result in large intra-class variations by grouping base classes without common properties. We instead sample $M$ meta-class sets by running $k$-means clustering algorithms with random seeds. Since the clustering algorithm often results in redundant meta-class sets despite the random seeds, we diversify the clustering results by randomly sampling $Q$-dimensional axis-aligned subspaces of class representation vectors. We obtain class embedding from the weights of the classification layer in a convolutional neural network, which is fine-tuned using noisy labels in the original class space.

While the clustering-based method is sufficiently good at mining reasonable meta-class sets, we can further optimize the configurations of meta-class sets by searching for their combinations. To achieve this, we oversample candidate meta-class sets, and search for the optimal subset using a search agent trained by reinforcement learning. Given a set of all candidate meta-class sets $\mathcal{P} = \{\mathcal{C}^m\}_{m=1}^M$, the search agent predicts a probability distribution over a binary selection variable $u_m$ for each candidate meta-class set. We train the agent by a policy gradient method, specifically REINFORCE rule [38], and iteratively update the parameters by the empirical approximation of the expected policy gradient, which is given by

$$\nabla_\theta \mathcal{J}(\theta) = \frac{1}{S} \sum_{s=1}^S \sum_{m=1}^M \nabla_\theta \log P(u_m^{(s)}; \theta)(\mathcal{R}^{(s)} - B), \tag{5}$$

where $S$ is the number of meta-class set combinations[2], $\theta$ is a model parameter of the search agent, $\mathcal{R}^{(s)}$ is the reward obtained by the $s$-th sample, and $B$ is the baseline injected to reduce the variance. Our main goal is to select the optimal collection of meta-class sets in terms of accuracy on the validation dataset, but we employ in-batch validation accuracy as the primary reward for training efficiency. In addition, we encourage the number of selected meta-class sets to be small in each combination by providing the negative reward in proportion to its size. Then, the total reward for the selection is given by

$$\mathcal{R} = \mathcal{R}_{\mathrm{acc}} - \alpha \sum_{i=1}^M u_i, \tag{6}$$

where $\mathcal{R}_{\mathrm{acc}}$ is the in-batch validation accuracy and $\alpha$ is a hyper-parameter balancing the two terms. Finally, we set the baseline to the average reward of the batch samples, *i.e.*, $B = \frac{1}{S} \sum_s \mathcal{R}^{(s)}$. At every $T$ epoch during training, we evaluate all the $S$ samples on the entire validation set and store the meta-class set combination with the highest accuracy. At the end of the training, the agent returns the best meta-class set combination among the stored ones. According to our empirical observations, the search cost by RL is just as much as the cost for training a classifier. Note that we employ a simple two-layer perceptron to optimize the agent, which is also helpful to reduce the computational complexity together with the in-batch validation strategy described above.

### 3.4 Discussions

In addition to the benefit of noise level reduction, the coarse-grained meta-class representation also brings several desirable characteristics. The meta-classes naturally introduce some inter-class relationships to the model and lead to better generalization performance by grouping multiple classes that potentially have shared information. Moreover, the representation learning based on meta-classes makes the trained model more robust to data-deficiency since each coarse-grained meta-class obviously contains more training examples compared to the original classes. As multiple meta-class sets construct a large number of class codewords by their Cartesian product, a small number of constituent classifiers are sufficient to recover the original class set and the proposed method can reduce the number of parameters. Finally, since the proposed method utilizes multiple constituent classifiers, it brings some ensemble effects and lead to accuracy improvement.

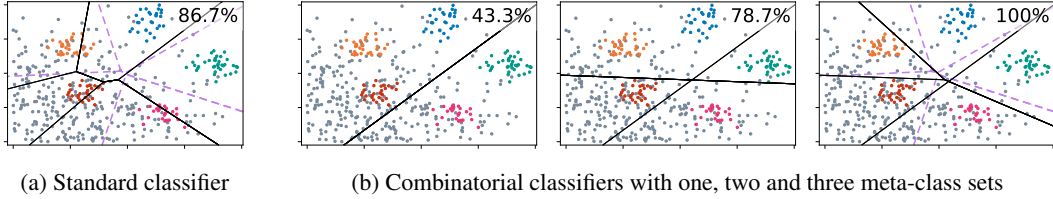

(a) Standard classifier  (b) Combinatorial classifiers with one, two and three meta-class sets

Figure 3: Sample results from (a) a standard classifier and (b) combinatorial classifiers on the examples from a 2D Gaussian mixture model with five components. The accuracy is shown at the top-right corner in each case. For the combinatorial classifiers, we gradually add meta-classifiers one-by-one. Gray dots correspond to noisy samples with random labels while purple dashed and black solid lines represent decision boundaries on clean and noisy datasets, respectively.

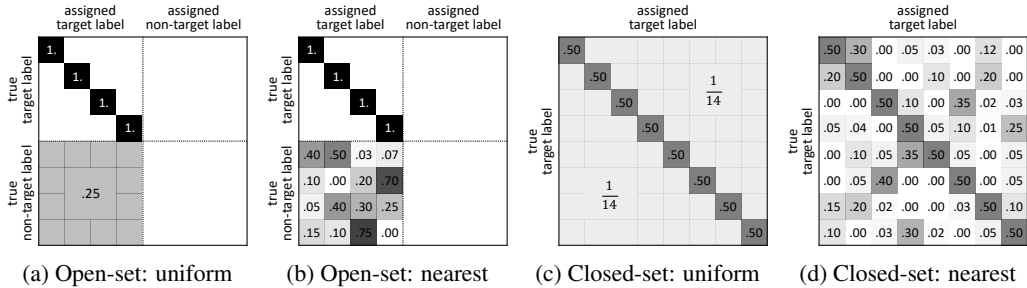

(a) Open-set: uniform  (b) Open-set: nearest  (c) Closed-set: uniform  (d) Closed-set: nearest

Figure 4: Example noise transition matrices with eight output classes. For open-set noise, four classes are used as target classes while the remaining four classes are reserved for noisy labels.

# 4 Experiments

## 4.1 Experiments on a Toy Set

To illustrate and visualize the effectiveness of the proposed method, we build a toy set that contains 150 clean samples drawn from a two-dimensional Gaussian mixture model with five components. To simulate a dataset with significant noise, we generate 300 noisy examples from a random Gaussian distribution with a larger variance and assign noisy labels selected from a uniform distribution. Figure 3 demonstrates the decision boundaries of a standard method and the proposed combinatorial classifiers with their accuracies. Our model is based on logistic regressors on three binary meta-class sets, which are gradually added one by one as shown in Figure 3b. With these results, we put emphases on the following three observations. First, the combinatorial classifier becomes capable of identifying all original classes as we add more meta-class sets. The decision boundaries and accuracies also illustrate the noise-robustness of the our method. Finally, the proposed technique requires fewer parameters (three weight vectors in the logistic regressors) than the standard classifier (five weight vectors corresponding to each class).

## 4.2 Evaluation on CUB-200

**Experimental settings**   We conduct a set of experiments on Caltech-UCSD Birds-200-2011 (CUB-200) dataset [39] with various noise settings. CUB-200 is a fine-grained classification benchmark with 200 bird species and contains ∼30 images per class in the training and validation sets. Note that CUB-200 is more natural and realistic compared to the simple datasets—MNIST and CIFAR—used for the evaluation of many previous methods.

We consider both open- and closed-set noise artificially injected to training examples. The open-set noise is created by giving one of the target labels to the images sampled from unseen categories. To simulate open-set noise, we use 100 (out of 200) classes as the target labels and the remaining 100 classes assume to be unknown. Noise level $\eta$ controls the ratio between clean and noisy examples. On the other hand, examples with the closed-set noise have wrong labels within the target classes and we use all 200 classes in CUB-200 as the target labels. For both types of noise, we use two label corruption schemes: uniform transition and nearest label transfer. The uniform transition injects label

Table 1: Accuracies [%] on CUB-200 with different levels of open-set noise.

| Methods | Clean dataset ($\eta = 0$) | Moderate noise level | | High noise level | |
|---|---|---|---|---|---|
| | | Uniform | Nearest | Uniform | Nearest |
| Standard | 80.57 ± 0.37 | 73.37 ± 0.34 | 77.14 ± 0.27 | 70.04 ± 0.71 | 75.45 ± 0.50 |
| Decoupling [6] | 79.32 ± 0.83 | 71.42 ± 0.70 | 76.07 ± 0.40 | 66.79 ± 0.44 | 74.80 ±0.46 |
| F-correction [19] | 80.66 ± 0.60 | 73.55 ± 0.70 | 77.03 ± 0.29 | 69.76 ± 0.59 | 75.52 ± 0.32 |
| S-model [18] | 80.75 ± 0.37 | 73.52 ± 0.47 | 77.13 ± 0.97 | 70.06 ± 0.65 | 75.59 ± 0.33 |
| MentorNet [7] | 80.39 ± 0.36 | 73.53 ± 0.56 | 77.27 ± 0.49 | 70.34 ± 0.42 | 75.75 ± 0.62 |
| $q$-loss ($q = 0.3$) [10] | 81.55 ± 0.52 | 75.04 ± 0.46 | 78.09 ± 0.40 | 71.84 ± 0.95 | 76.32 ± 0.65 |
| $q$-loss ($q = 0.5$) [10] | 82.19 ± 0.58 | 77.51 ± 0.53 | 78.58 ± 0.46 | 75.40 ± 0.66 | 76.47 ± 0.34 |
| $q$-loss ($q = 0.8$) [10] | 75.15 ± 1.96 | 71.02 ± 1.68 | 47.40 ± 1.16 | 67.16 ± 1.29 | 32.60 ± 2.38 |
| Co-teaching [11] | 80.90 ± 0.13 | 75.37 ± 0.54 | 77.41 ± 0.36 | 74.02 ± 0.29 | 75.57 ± 0.22 |
| CombCls | 82.80 ± 0.36 | 79.38 ± 0.83 | 79.28 ± 0.52 | 79.19 ± 0.29 | 77.95 ± 0.53 |
| CombCls+Co-teaching | **82.86 ± 0.25** | **79.93 ± 0.44** | **80.22 ± 0.31** | **80.50 ± 0.61** | **78.26 ± 0.43** |

Table 2: Accuracies [%] on CUB-200 with different levels of closed-set noise.

| Methods | Clean dataset ($\eta = 0$) | Moderate noise level | | High noise level | |
|---|---|---|---|---|---|
| | | Uniform | Nearest | Uniform | Nearest |
| Standard | 79.58 ± 0.18 | 63.65 ± 0.26 | 65.21 ± 0.42 | 42.35 ± 0.50 | 47.70 ± 0.41 |
| Decoupling [6] | 77.79 ± 0.23 | 62.52 ± 0.23 | 66.24 ± 0.53 | 43.91 ± 0.52 | 51.92 ± 0.18 |
| F-correction [19] | 80.01 ± 0.42 | 63.81 ± 0.16 | 64.69 ± 0.21 | 42.23 ± 0.54 | 48.00 ± 0.46 |
| S-model [18] | 79.42 ± 0.27 | 63.08 ± 0.74 | 64.90 ± 0.29 | 42.17 ± 0.70 | 48.01 ± 0.47 |
| MentorNet [7] | 79.78 ± 0.20 | 68.03 ± 0.32 | 65.49 ± 0.14 | 47.74 ± 1.64 | 48.25 ± 0.39 |
| $q$-loss ($q = 0.3$) [10] | 80.41 ± 0.36 | 68.52 ± 0.51 | 66.34 ± 0.25 | 53.18 ± 0.49 | 49.30 ± 0.35 |
| $q$-loss ($q = 0.5$) [10] | 80.76 ± 0.38 | 75.24 ± 0.31 | 67.49 ± 0.56 | 60.89 ± 0.32 | 49.28 ± 0.57 |
| $q$-loss ($q = 0.8$) [10] | 40.70 ± 2.25 | 29.31 ± 1.14 | 24.98 ± 1.61 | 17.67 ± 1.06 | 15.95 ± 0.65 |
| Co-teaching [11] | 79.74 ± 0.14 | 68.21 ± 0.35 | 66.24 ± 0.30 | 52.72 ± 0.56 | 49.81 ± 0.19 |
| CombCls | 81.36 ± 0.23 | 71.75 ± 0.24 | 68.35 ± 0.35 | 51.90 ± 0.35 | 52.00 ± 0.22 |
| CombCls+Co-teaching | **81.52 ± 0.47** | **75.30 ± 0.10** | **70.46 ± 0.31** | **62.77 ± 0.66** | **52.49 ± 0.79** |

noise by selecting a wrong label uniformly while the nearest label transfer determines the label of a noisy example using the nearest example with a different label to simulate confusions between visually similar classes. For the nearest neighbor search, we employ the features of examples in a pretrained network on the clean dataset. For both noise types, we test moderate and high noise levels ($\eta = 0.25$ and $\eta = 0.50$). Figure 4 presents sample noise transition matrices for all cases.

We use ResNet-50 as the backbone network for all the methods and initialize the parameters of the feature extractor using the pretrained weights on ImageNet [40] while the classification layer(s) are initialized randomly. The entire network is fine-tuned for 40 epochs by a mini-batch stochastic gradient descent method with batch size of 32, momentum of 0.9 and weight decaying factor of $5 \times 10^{-4}$. The initial learning rate is 0.01 and decayed by a factor of 0.1 at epoch 20 and 30. For combinatorial classification, we use 100 binary meta-class sets ($K' = 2$) generated by performing $k$-means clustering with $Q = 50$. All models are evaluated on clean test sets with five independent runs. We report the best test accuracy across epochs for all models for comparisons since the learning curves of individual methods may be different and reporting accuracies at a particular epoch may be unfair. However, we also note that our approach still outperforms others even when fixing the number of epochs in a wide range.

**Results** The proposed combinatorial classification (CombCls) is compared with the following state-of-the-art methods including Decoupling [6], F-correction [19], S-model [18], MentorNet [7], $q$-loss [10] and Co-teaching [11], in addition to an ordinary flat classifier (Standard).

Table 1 and 2 present results on CUB-200 dataset in the presence of open- and closed-set noises, respectively. We first observe that CombCls outperforms Standard on noise-free setting ($\eta = 0$) in both cases. This is partly because our combinatorial classifier learns useful information for recognition by modeling inter-class relationships and exploits ensemble effects during inference as discussed in Section 3.4. These results imply that our method is also useful regardless of noise-level and achieves outstanding classification accuracy. Moreover, the proposed algorithm identifies a compact model compared to the other methods. The baseline models use $K$ weight vectors in their classification layers, where each vector corresponds to a base class; the baselines have 100 and 200

Table 3: Results of ablation studies. Accuracies [%] of (left) combinatorial classifier trained on highly noisy datasets with meta-class sets generated from datasets with different levels of uniform noise and (right) standard classifier with feature extractor of CombCls in various noise configurations.

| Dataset used for meta-class set generation | Open-set | Closed-set |
|---|---|---|
| Clean | 77.82 | 45.40 |
| Moderate noise level | 77.58 | 50.22 |
| High noise level | 79.19 | 51.90 |

| | | Standard | Standard +CombFeat |
|---|---|---|---|
| Open-set | Uniform | 70.04 | 78.48 |
| | Nearest | 75.45 | 77.40 |
| Closed-set | Uniform | 42.35 | 53.83 |
| | Nearest | 47.70 | 52.52 |

Table 4: Results of combinatorial classification using different meta-class set configurations on the datasets with high noise level. Acc. means accuracy [%] and Param. is ratio of model parameters in each method with respect to that of the Standard.

| | | Open-set noises | | | | Closed-set noises | | | |
|---|---|---|---|---|---|---|---|---|---|
| | | Uniform | | Nearest | | Unifrom | | Nearest | |
| Methods | Meta-class set | Acc. | Param. | Acc. | Param. | Acc. | Param. | Acc. | Param. |
| Standard | N/A | 70.04 | 1.00 | 75.45 | 1.00 | 42.35 | 1.00 | 42.84 | 1.00 |
| CombCls | Random | 78.66 | 1.00 | 76.75 | 1.00 | 46.98 | 0.50 | 48.55 | 0.50 |
| | Clustering | 79.19 | 1.00 | 77.95 | 1.00 | 51.90 | 0.50 | 52.00 | 0.50 |
| | Clustering+Search | **79.98** | **0.42** | **78.35** | **0.44** | **54.52** | **0.41** | **52.43** | **0.29** |

weight vectors for open- and closed-set noise cases. Note that CombCls consists of $M(=100)$ binary classifiers in both cases and saves memory substantially.

Our algorithm outperforms other methods in the presence of noise and the accuracy gain is even larger than noise-free case. It achieves the state-of-the-art accuracy in most settings. Note that $q$-loss with the optimal $q$ value is better than our method when uniform closed-set noise is injected. This is mainly because such problem configurations are aligned well to the assumption behind $q$-loss algorithm [10]. However, real noise distributions are unlikely to follow uniform distributions. For instances, we observe significantly more open-set noise than closed-set one in a real-world noisy dataset as shown in Figure 1. Moreover, the performance of $q$-loss highly depends on the choice of $q$ and an inappropriate choice of $q$ value degrades performance significantly because the theoretical noise-robustness and training stability vary with respect to $q$ values; the optimal $q$ value differs across datasets, *e.g.*, $q = 0.5$ for CUB-200 and $q = 0.3$ for WebVision (shown in Section 4.3). In contrast, our algorithm reduces the level of noise effectively regardless of hyper-parameters by introducing coarse-grained meta-class sets. Another observation is that our approach is unique and complementary to other methods. As a result, it is straightforward to further improve accuracy by combining both our method and Co-teaching [11] (CombCls+Co-teaching). Note that Co-teaching trains two networks to filter out noisy examples by cross referencing each other, which employs a completely different approach from the proposed one to tackles label noise. Noticeably, CombCls+Co-teaching achieves the best accuracy through the collaboration in all noise configurations and almost recovers the accuracy of Standard on the clean dataset in Table 1.

For further understanding, we train the proposed classifier on the datasets with high noise level while pretraining the network for the clustering-based meta-class set generation on another dataset with different noise levels. Interestingly, using the dataset with high noise level in the meta-class sets generation gives higher accuracy compared to the cleaner datasets as shown in Table 3(left). This implies that the meta-class sets generated from a noisy dataset reflect the noise distribution and help the combinatorial classifiers generalize better on the noisy dataset. We observe similar tendencies with other combinations of noise levels, where the meta-class sets generated with the same noise distribution result in higher accuracies. Also, we construct a Standard network and initialize its feature extractor using a trained CombCls model. Then, we fine-tune its classification layer while the weights of the feature extractor are fixed. This network (Standard+CombFeat) is compared to Standard in Table 3(right). Using the feature extractor of the combinatorial classifier, the accuracy of Standard is improved with a significant margin in all noise settings. This signifies that the proposed model learns a noise-robust feature extractor.

Next, we evaluate the proposed method with different meta-class set configurations and show the results in Table 4. We compare randomly configured meta-class sets (Random) with the proposed clustering based ones (Clustering). In addition, we also evaluate performance of the meta-class sets

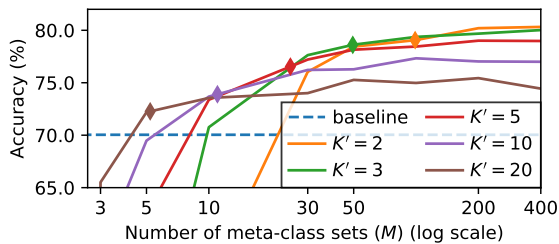

Figure 5: Accuracy [%] of combinatorial classifiers with open-set uniform noise ($\eta = 0.5$) by varying $K'$ and $M$.

Table 5: Results on WebVision.

| Methods | Acc. [%] |
|---|---|
| Standard | 79.82 |
| Decoupling [6] | 79.38 |
| F-correction [19] | 80.96 |
| S-model [18] | 81.36 |
| MentorNet [7] | 80.46 |
| $q$-loss ($q = 0.3$) [10] | 82.18 |
| Coteaching [11] | 83.06 |
| CombCls | 83.14 |
| CombCls+Search | 83.26 |
| CombCls+Coteaching | **84.14** |

search method (Clustering+Search) proposed in Section 3.3. For the meta-class set search, we reserve a half of training set as the validation set to train the search agent, compute the in-batch accuracy with 32 images, and sample 100 meta-class set combinations per batch. The number of candidate meta-class sets is 600 and $\alpha$ in Eq. (6) is set to $3 \times 10^{-4}$. Note that the search process does not require any clean data since the agent is trained on the noisy held-out data extracted from the training set. After the search process, we retrain the combinatorial classifier with the entire training set including the validation data. As shown in Table 4, the combinatorial classifier outperforms the baseline (Standard) even with randomly constructed meta-class sets. Our clustering-based algorithm brings additional improvement while employing the meta-class set search technique boosts the accuracy even further. Note that the meta-class sets configured by the search agent not only improves the accuracy of CombCls but also reduces the number of parameters as the meta-class sets are optimized to maximize the accuracy and minimize the number of meta-class sets.

Finally, we investigate the effects of $K'$ and $M$ in our method. Figure 5 shows the effects of $K'$ and $M$ on CUB-200 with open-set uniform noise ($\eta = 0.5$). Our models outperform the baseline even with a fairly small number of meta-class sets regardless of $K'$. In particular, the model with $K' = 2$ reduces noise level most effectively and achieve the best accuracy among the ones with the same number of parameters, which are depicted by diamond markers in the plot. We observe the same tendency in the experiments with other noise settings.

## 4.3 Experiments on WebVision

We also conduct experiments on a real-world noisy benchmark, WebVision [4]. This dataset is constructed by collecting 2.4 million web images retrieved from Flickr and Google using manually defined queries related to 1,000 ImageNet classes. While the training set includes significant amount of noise, the benchmark provides a clean validation set for evaluation. We use a subset of WebVision dataset for our experiment, which contains all images from 100 randomly sampled classes. The experimental settings are identical to the ones described in the previous section except for the optimization parameters; we adopt the parameters of the ImageNet training setting in [41].

Table 5 presents accuracies of all compared methods and the proposed model shows competitive performance. As in the experiments on CUB-200, our method benefits from the combination with Co-teaching and achieves the best accuracy. We also find that applying the meta-class sets search makes additions accuracy gain. Moreover, it reduces the model complexity and uses only 64% of the parameters in the classification layers compared to the baselines and the proposed model without the meta-class sets search.

## 5 Conclusion

We proposed a novel classification framework, which constructs multiple classifiers over heterogeneous coarse-grained meta-class sets and perform combinatorial inferences using their predictions to identify a target label in the original class space. Our method is particularly beneficial when the dataset contains label noise since the use of the coarse-grained meta-class representations reduces noise level naturally. We also introduced meta-class set search techniques based on clustering and reinforcement learning. The extensive experiments on the datasets with artificial and real-world noise demonstrated the effectiveness of the proposed method in terms of accuracy and efficiency.

**Acknowledgments**

This work is partly supported by Google AI Focused Research Award and Korean ICT R&D programs of the MSIP/IITP grant [2017-0-01778, 2017-0-01780].

## Footnotes

[1]Refer to Section 4.2 for noise injection configurations.

[2]Each combination is composed of the meta-class sets and each of the meta-class sets is selected by $u_m$.

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
