[Reviews · NeurIPS 2019]

Reviewer 1



The paper proposes to use multiple meta classes, each of which which are defined to be partitions of the original classes, to create a classifier that is more robust to label noise. The idea is that, while some of the original labels may be incorrect, they are more likely to be correct in the new meta classes. Furthermore, given enough overlapping meta classes, it is possible to infer the original class by meta class membership. In practice, the authors use meta classes that are binary partitions of the original class space, which are constructed by clustering on subsets of features from a classifier trained on the original noisy labels. This is a nice idea, and experiments show that it works well. Originality: the idea here is novel (as far as I know) and innovative, going beyond an incremental contribution. Related work is adequately discussed. Clarity: overall, the paper is quite clear and provides a sufficient level of detail. Some minor language errors remain, but very readable for the most part. Significance: interesting idea, advances the state of the art, likely to be interesting for others working in the area.

Reviewer 2



The combinatorial (meta- or super-class) idea is interesting: it is reasonable and one easily expects to work well. In terms of related work, I suggest add 2 related papers. One is ECOC (Solving Multiclass Learning Problems via Error-Correcting Output Codes, JAIR 1995), which is a classic combinatorial method for classification. The other one is PENCIL (Probabilistic End-to-end Noise Correction for Learning with Noisy Labels, CVPR 2019), which is a novel noise handling method. With regard to the method, the proposed probabilistic way to decipher class from meta-class is simple. Does it have any guarantee (especially with under different hyper-parameters like k' and M)? It is possible that the mathematical underpinning of ECOC could be useful. For clustering, the current description is pretty terse. For example, I suggest the author(s) may make use of a supplementary material to provide more details (this submission did not have a supplementary material, though). Finally, why the experiments on WebVision are a small scale / downsampled one? In fact, this is the most important experiment (because it is real-world data). I expect to see how the CombCls method works in reality. And, I suggest ablation studies in terms of hyper parameters, especially K' and M. -- The author response addressed most of my questions. The large scale one is missing -- it is reasonable because there may not be enough time to run a large scale webvision experiment during the short rebuttal period. However, I suggest that the author(s) test this method in real-world large scale problems.

Reviewer 3



Originality: The ideas in the paper are quite original. Related work has been cited well. Quality: The work is technically sound. The conclusions are supported by the experiments. Clarity: Please give more details of the reinforcement learning task used for learning optimal collection meta class sets. Some intuition on what makes an optimal collection should be provided. Significance: The improvements are modest for the WebVision dataset. More empirical results would be required to assess the significance. --- Post rebuttal response: The authors responded to my concerns adequately. After looking at the comments of other reviewers and the author's response, I'd like to change my score to 7.

[Author Response · NeurIPS 2019]

We appreciate positive and constructive comments, and address the main concerns raised by the reviewers below.

**Effects of $K'$ and $M$ [R4]**  Figure 1 shows the effects of $K'$ and $M$ on CUB-200 with open-set uniform noise ($\eta = 0.5$). Our models outperform the baseline even with a fairly small number of meta-class sets regardless of $K'$. In particular, the model with $K' = 2$ works best among the ones with the same number of parameters, which are depicted by a diamond marker on each line plot, since it reduces noise level most effectively. We observe the same tendency in the experiments with other noise settings.

Figure 1: Accuracy by varying $K'$ and $M$.

**Relation to ECOC [R4]**  Unlike ECOC that constructs codewords deterministically, it is not straightforward to show a theoretical guarantee of our method based on an iterative clustering algorithm in a high-dimensional latent feature space. However, our approach increases the number of partitions exponentially by simply adding meta-class sets, which makes a small number of meta-class sets ($M = 20$) large enough to decipher all 200 classes on CUB-200 through a combination of binary meta-class sets ($K = 2$). It is a valuable suggestion to adopt the intuition behind ECOC for meta-class set configuration. It effectively decreases the number of meta-class sets to 15 from 20 for deciphering in practice. Moreover, when we employ the column separation idea proposed in the ECOC paper and reduce the correlation between meta-class sets, our approach shows additional accuracy gains by approximately 1% point in average on CUB-200.

**Optimality of meta-class sets [R5]**  The ideal meta-class sets should be compact and uncorrelated, and these properties affect the final accuracy of our algorithm. Our reward function (Eq. (6) of the main paper) guides to identify the meta-class sets with such desirable characteristics while there is no direct supervision for correlation between meta-class sets. However, it turns out that the meta-class sets given by our RL-based method are significantly less correlated compared to the random selection. We will consider a more direct feedback based on the measured correlation while a simple approach has already been discussed in "Relation to ECOC [R5]". Thanks for the insightful comment.

**Issues related to RL [R1]**  Our meta-class set search algorithm relies on a noisy validation set held out from the training set, which is also adopted in the experiments on WebVision. The search cost by RL is just as much as the cost for training a classifier. To reduce the computational complexity, we employ a simple two-layer perceptron and optimize the agent based on the in-batch validation accuracy instead of computing the accuracy on the entire validation set.

**Ablation study on meta-class set construction methods [R1]**  As shown in Table 1, the combinatorial classifier outperforms the baseline (Standard) even with randomly constructed meta-class sets (Random). Our clustering-based algorithm (Clustering) brings additional improvement especially on the datasets with high noise level while employing the meta-class set search technique (Clustering+RL_Search) boosts the accuracy even further.

Table 1: Accuracies [%] of baseline and proposed models with different meta-class set configurations on CUB-200.

| | Clean | Open-Uniform | | Open-Nearest | | Clean | Closed-Uniform | | Closed-Nearest | |
|---|---|---|---|---|---|---|---|---|---|---|
| | $\eta = 0$ | $\eta = 0.25$ | $\eta = 0.50$ | $\eta = 0.25$ | $\eta = 0.50$ | $\eta = 0$ | $\eta = 0.25$ | $\eta = 0.50$ | $\eta = 0.25$ | $\eta = 0.50$ |
| Standard | 80.57 | 73.37 | 70.04 | 77.14 | 75.45 | 79.58 | 63.65 | 42.35 | 65.21 | 47.70 |
| Random | 82.05 | **80.05** | 78.66 | 78.74 | 76.75 | 79.82 | 69.36 | 46.98 | 67.12 | 48.55 |
| Clustering | 82.80 | 79.38 | 79.19 | 79.28 | 77.95 | 81.36 | 71.75 | 51.90 | 68.35 | 52.00 |
| Clustering+RL_Search | **83.12** | 79.72 | **79.98** | **79.36** | **78.35** | **81.62** | **72.43** | **54.52** | **68.79** | **52.43** |

**Use of pretrained ImageNet [R1]**  Although we partly agree to the concern about the use of the pretrained network on ImageNet, we also believe that the pretrained model is a commodity and is widely used for the initialization of many CNNs. Also, our experiment on WebVision does not rely on the pretrained model but the CNNs trained from scratch; it illustrates the representation learning capability of the proposed method.

**Results on CIFAR-100 [R1]**  Table 2 presents the results from the methods tested on CIFAR-100 with the ResNet-50 backbone model, where the proposed approach (CombCls) achieves the largest accuracy gains with respect to the baseline (Standard) and a combination with Co-teaching (CombCls+) improves accuracies further.

Table 2: Results on CIFAR-100 with open-set noise ($\eta = 0.25$).

| | Uniform | Nearest |
|---|---|---|
| Standard | 73.51 | 73.88 |
| $q$-loss | 73.91 | 74.41 |
| Co-teaching | 75.84 | 76.30 |
| CombCls | 76.30 | 76.69 |
| CombCls+ | **78.57** | **78.39** |

**Qualitative examples in meta-class sets [R1]**  The binary separations of input images given by meta-class sets look reasonable; each meta-class captures either one or more attributes, related classes, or common visual appearances, which are crucial cues to identify the fine-grained categories. It is difficult to include specific cases in the rebuttal, but we will present interesting examples in the supplementary file if our paper is accepted.

**Final vs. best accuracy [R1]**  The results from all algorithms are given by the models identified using a clean validation set. This is because the learning curves of individual methods may be different and reporting accuracies at a particular epoch may be unfair. We also note that our approach still outperforms others even when we fix the number of epochs in a wide range for comparisons. We will present more detailed results if our paper is accepted.

**Others [All]**  We will supplement the missing details and results in the final manuscript if our paper is accepted.

[Meta-Review · NeurIPS 2019]

This paper proposed a completely new way for learning with noisy labels: it smartly construct certain meta classes and learn classifiers to predict meta-class labels; the predictions of the original base-class labels can be inferred by combining the predictions on multiple meta-class spaces (each has several meta classes). As a result of this algorithm design, the noise level of label noise is empirically shown to be reduced in the meta-class space compared with the original base-class space (though this benefit is not theoretically guaranteed). The intuition behind the idea is that multiple base classes collapse to a single meta class and then the label noise within the same meta-class vanishes. The clarity, the novelty, and the significance are all above the corresponding thresholds of NeurIPS and thus it should clearly be accepted. The problem under consideration is of practical interests and may have huge impacts to our daily life (as mentioned in the intro, noisy labels are everywhere in the wild). This paper manipulates the output representation (which is the fitting target); it is slightly similar to label correction, but the gap between them is significant enough to make this paper stand for a new direction in label-noise learning. In order to address the broader audience in NeurIPS, I offer my quick thoughts on the paper (I didn't carefully check the full paper due to limited time): A. The title is too short and not sufficiently informative---the core concept "meta class" doesn't appear in the title now! In principle I shouldn't affect your choice of the title too much, since the title is most important for a paper, I strongly suggest you to consider a title that can reflect the intuition (i.e., multiple base classes collapse to a single meta class and then the label noise within the same meta-class vanishes). B. In the literature review, sample selection/reweighting methods and label correction methods are combined. While both of those two directions try to identify good data from noisy data, the former simply drops possibly bad data but the latter still tries to fix the labels of those bad data. This difference should be mentioned in the sections of introduction and related work. C. Two related papers following Co-teaching should be cited: Co-teaching+ (entitled "How does disagreement help generalization against label corruption?") and Pumpout (entitled "Pumpout: A meta approach to robust deep learning with noisy labels"). They went along the line of sample selection. D. There are some typos and grammatical issues. Please check the English, carefully, once more. [This meta-review was reviewed and revised by the Program Chairs]